# Electrophoretic Determination of Symmetric and Asymmetric Dimethylarginine in Human Blood Plasma with Whole Capillary Sample Injection

**DOI:** 10.3390/ijms22062970

**Published:** 2021-03-15

**Authors:** Petr Tůma, Blanka Sommerová, Dušan Koval, François Couderc

**Affiliations:** 1Department of Hygiene, Third Faculty of Medicine, Charles University, Ruská 87, 100 00 Prague 10, Czech Republic; blanka.sommerova@lf3.cuni.cz; 2Institute of Organic Chemistry and Biochemistry, The Czech Academy of Sciences, Flemingovo n. 2, 166 10 Prague 6, Czech Republic; koval@uochb.cas.cz; 3Laboratoire des IMRCP UMR 5623, University Toulouse III-Paul Sabatier, 31062 Toulouse, France; couderc@chimie.ups-tlse.fr

**Keywords:** amino acid, capillary coating, capillary electrophoresis, clinical analysis, contactless conductivity detection, dimethylarginines, sample treatment, stacking

## Abstract

Asymmetric and symmetric dimethylarginines are toxic non-coded amino acids. They are formed by post-translational modifications and play multifunctional roles in some human diseases. Their determination in human blood plasma is performed using capillary electrophoresis with contactless conductivity detection. The separations are performed in a capillary covered with covalently bonded PAMAPTAC polymer, which generates anionic electroosmotic flow and the separation takes place in the counter-current regime. The background electrolyte is a 750 mM aqueous solution of acetic acid with pH 2.45. The plasma samples for analysis are treated by the addition of acetonitrile and injected into the capillary in a large volume, reaching 94.5% of the total volume of the capillary, and subsequently subjected to electrophoretic stacking. The attained LODs are 16 nm for ADMA and 22 nM for SDMA. The electrophoretic resolution of both isomers has a value of 5.3. The developed method is sufficiently sensitive for the determination of plasmatic levels of ADMA and SDMA. The determination does not require derivatization and the individual steps in the electrophoretic stacking are fully automated. The determined plasmatic levels for healthy individuals vary in the range 0.36–0.62 µM for ADMA and 0.32–0.70 µM for SDMA.

## 1. Introduction

N-methylation of the amino acid L-arginine in proteins is a common post-translation modification, which substantially extends the set of functional proteins in the individual cells. Methylation of the N-terminal atoms of the L-arginine side chain is catalysed by a group of enzymes designated as protein arginine methyltransferases (PRMTs), which employ S-adenosylmethionine as a methyl donor [1]. There are two groups of PRMTs: In the first step, Class I catalyses methylation of one terminal N atom to form monomethylarginine (MMA) and subsequent methylation leads to the formation of asymmetric dimethylarginine (ADMA). On the other hand, the PRMTs of Class II synthesise MMA in the first phase, but the second methylation leads to the formation of symmetric dimethylarginine (SDMA), see Figure 1. Because of their high basicity, arginine methylated proteins react strongly with nucleic acids and participate in processes such as transcription, translation, etc. The biological function, analysis and identification of arginine methylated proteins is summarised in general reviews [2,3].

Methylation of proteins is an irreversible process and the hydrolysis of proteins in the context of their turnover leads to the release of unbounded ADMA, SDMA and a small amount of MMA into the cytoplasm. A human being produces approximately 300 µmol ADMA per day; 90% of this amount is intracellularly degraded by enzyme dimethylarginine dimethylaminohydrolase (DDAH) to form citrulline and dimethylamine and the remaining 10% is transported to the blood and excreted in the urine in unaltered form [4,5]. Analogously, DDAH also decomposes MMA to methylamine and citrulline. However, SDMA is not degraded through DDAH. This explains why the plasmatic levels of ADMA and SDMA are approximately identical in spite of the fact that primary methylation forms more ADMA. The final ADMA, SDMA, MMA and dimethylamine are removed from the blood circulation system by renal excretion into the urine [6,7].

The pathophysiological role of ADMA is described in detail in two reviews [8,9], and will be mentioned only briefly here. It was demonstrated at the beginning of the 1990s that the accumulation of methylarginines in patients with end-stage renal disease is manifested by hypertension and immune dysfunction [6]. The explanation consists of the inhibition of isoforms of nitric oxide synthase (NOS), which catalyses the synthesis of nitric oxide (NO) from arginine, where citrulline is a side product of this reaction. ADMA and MMA act as competitive inhibitors of NOS, while SDMA does not act on NOS in this way. NO has a number of physiological functions in the vascular, immunity and nervous systems. A high ADMA level has primarily been connected with the development of cardiovascular diseases through inhibition of the synthesis of endothelial NO [10]. Endothelial NO is an important endogenic vasodilator and also inhibits a number of processes causing atherosclerosis, such as aggregation of platelets, proliferation of smooth vessel muscle cells and adhesion of inflammatory cells to the vessel walls [11,12,13]. Elevated plasmatic ADMA levels, as a risk factor in cardiovascular disease, occur in cases of hypercholesterolaemia [14], insulin resistance and diabetes mellitus [15], chronic kidney disease [16,17] and hypertension [18,19]. The difference between the elevated plasmatic level of ADMA in the risk group compared to healthy individuals is not large. It is necessary to realise that ADMA is synthesised and acts in the cytoplasm, where its concentration is an order of magnitude larger than in the blood plasma [20]. SDMA is not a direct inhibitor of NOS, but only competes with arginine for transport directed by the cationic amino acid transporter, which reduces the availability of arginine as a substrate for NOS and thus participates to a certain extent in the reduction synthesis of NO [21].

The majority of clinical determinations of ADMA and SDMA are performed in the blood plasma, where the levels of both modified amino acids are at submicromolar levels, while the concentrations of the other amino acids are 100 to 1000 times higher. This fact places very high demands on the employed analytical technique. ADMA and SDMA have been determined using ion-exchange liquid chromatography with post-column derivatisation, reversed-phase HPLC with direct UV detection at 200 nm or far more often by HPLC after pre-column derivatisation. In this case, the derivatisation reagent employed are o-phthaldialdehyde (OPA) [22], naphthalene-2,3-dicarboxaldehyde (NDA) [23], 6-aminoquinolyl-N-hydroxysuccinimidyl carbamate (AQC) [24] and 4-fluoro-7-nitro-2,1,3-benzoxadiazole (NBDF) [25]. The main disadvantage of HPLC, in addition to the long analysis time, is the demanding preparation of the plasma, including solid-phase extraction (SPE) [26]. On the other hand, for methods based on mass spectrometry (MS), single-step precipitation of the plasmatic proteins is usually sufficient, and the analysis time is thus shortened. The most commonly used method employs a combination of reversed phase liquid chromatography with tandem mass spectrometry (LC-MS/MS) without derivatisation [27] or after derivatisation by butyl ester [28] or a diethylpyrocarbonate-based derivatisation [29]. Methods of hydrophilic interaction liquid chromatography (HILIC) with MS without sample derivatisation are also known. In addition, a combination of gas chromatography with mass spectrometry (GC-MS) has been employed following derivatisation with pentafluoropropionyl (PFP), as summarised in general surveys [8,30,31,32,33,34,35]. A validated enzyme-linked immunosorbent assay (ELISA) has also been developed and is useful for the analysis of extensive sets of samples. However, ELISA yields significantly higher values (approximately twice as large) compared with HPLC or LC-MS [36].

The determination of plasmatic levels of methylarginines is a great challenge for capillary electrophoresis (CE). The technique of capillary zone electrophoresis (CZE) combined with laser-induced fluorescence (LIF) after pre-column derivatisation by fluorescein isothiocyanate (FITC) could be used to determine elevated levels of ADMA in samples of hemodialysed serum [37]. The CZE-UV determination of both derivatives without derivatisation is connected with double off-line laboratory preconcentration based on evaporation of the plasma sample to dryness [38,39]; the method was subsequently also used for the determination of the overall ADMA and SDMA contents in blood after total hydrolysis of the plasmatic proteins [40]. CZE-UV determination of the unbonded plasmatic levels was improved through the introduction of an on-line sharpening technique (stacking), for which single-step off-line plasma preconcentration following evaporation to dryness was sufficient [41,42]. The plasmatic levels were determined using a combination of CZE with tandem-MS improvement through the introduction of on-line sample stacking [43]. The methylarginines derivatised by naphthalene-2,3-dicarboxaldehyde (NDA) were separated by CZE and microchip electrophoresis (MCE) with LIF in model samples [44]. The CZE method was further extended by the introduction of heat-assisted extraction of methylarginines and employed for plasma analysis [45]. In this paper, we want to present a quick sample preparation and separation of ADMA and SDMA in plasma using CE with contactless conductivity detection (C^4^D).

Nonselective conductivity detection in the form of C^4^D is employed for detection of methylarginines, which exhibit only a very weak response in UV photometry at low wave lengths around 200 nm [46]. Methylarginines appear in C^4^D in BGE based on AcOH as negative peaks because the mobility of their cations is less than the mobility of the BGE co-ions, the highly mobile H_3_O^+^ ions [47]. A further advantage of C^4^D is its compatibility with a capillary with an inner diameter (ID) of ≤25 µm. The sensitivity of C^4^D does not decrease with a decrease in the ID as is true for UV-Vis photometry in CE. The sensitivity of C^4^D is controlled by the impedance of solution in the separation capillary; for a small ID, it is necessary to use BGE with high conductivity, and is also controlled primarily by the frequency of the exciting signal. It was demonstrated in previous studies [48] that a sensitivity at the level of 10^−6^ M can be attained without the use of stacking, even in thin capillaries with an ID ≤25 µm, which are characterised by a high separation efficiency, as a necessary condition for successful separation of complicated clinical samples.

## 2. Results and Discussions

### 2.1. Electrophoretic Stacking with Whole Capillary Sample Injection

Separation of methylarginines as basic amino acids is performed in AcOH solutions. The tested concentration range varied from 250 to 2000 mM AcOH and the optimum separation of ADMA and SDMA from the other components of the plasma was obtained in 750 mM AcOH (pH 2.45), which was used as BGE in all the experiments described below. Arginine and methylarginines are amino acids with two basic groups (an aminogroup with pKa 9.09, a biguanidine group in the side chain with pKa 13.2) and an acidic carboxyl group with pKa 2.18 (dissociation constant for arginine). At pH 2.45, both the basic groups are protonated, the carboxyl is mostly deprotonated, the overall charge on the amino acid is positive and methylarginines migrate as cations in the cationic direction. Simultaneously, the adsorption of cations of methylarginines on the inner surface of the capillary covered with cationic PAMAPTAC is suppressed.

As the plasmatic levels of ADMA and SDMA lie at submicromolar concentration levels, electrophoretic stacking was proposed, based on large-volume sample injection. Here, the entire capillary was filled with the sample. Successful whole capillary injection is connected with several specific features. A short BGE zone must be drawn into the capillary—otherwise the stacking results are worse (Figure A1 in Appendix A). During stacking, it is necessary to force the residual matrix out of the separation capillary by the use of EOF. In principle, the analyte is not lost during forcing out of the matrix using EOF as, if the analyte is to reach the detector, its electrophoretic speed must be greater than the speed of the EOF acting in the opposite direction and both are controlled by the intensity of the separation field. Contrary, the forcing out of the matrix by an external pressure is accompanied by sample losses, as was demonstrated in our previous work [49]. With PAMAPTAC coating, the magnitude of the EOF is controlled by the content of cationic APTAC components in the polymerisation mixture, where 6% PAMAPTAC coating was found suitable for the separation of methylarginines [50].

A recording of the separation of a model mixture of methylarginines at the 0.1 µM concentration level, including monitoring of the electric current, is summarised in Figure 2. Before commencing the separation, the sample, which has greater conductivity than the BGE, is located at the position of the detector. After turning on the high voltage, substantial changes occur in the conductivity of the solution, caused by the stacking, and subsequently a broad peak of Na^+^ is recorded. Finally, C^4^D records the baseline BGE with the negative peaks of the methylarginines. The electrical current is firstly high during stacking, corresponding to the capillary filled with the sample, after which it decreases to a minimum of about 1 µA, followed by a rapid increase as the residual ACN is forced out of the capillary. Finally, it attains a constant value when the whole capillary is filled with BGE. The state around the minimum is critical for maintaining the separation. However, no interruption occurred in the separation in the completed experiments with forcing out the residual ACN from the capillary using EOF.

The peaks of the methylarginines at a concentration level of 0.1 µM are clearly distinguished from the baseline noise; the signal/noise ratio is 101 for MMA, 95 for ADMA and 90 for SDMA. The electrophoretic resolution *R* of the neighbouring peaks of MMA and ADMA is 7.7 and equals 5.4 for the ADMA/SDMA pair. This high electrophoretic resolution of the individual methylarginines, with very close values of the electrophoretic mobilities, is attained through the opposite-direction action of EOF. The speed of the analyte in the capillary is proportional to the difference between the electrophoretic mobility and the EOF mobility, resulting in a slow rate of migration within the capillary. The electrophoretic separation time is long, contributing to mutual separation of the analytes with similar mobility values, as was demonstrated in our previous paper [50].

### 2.2. Determining ADMA and SDMA in Human Plasma

The developed stacking was applied to the determination of methylarginines in blood plasma treated by the addition of ACN in a ratio 1:3 and once again 94.5% of the capillary is filled with the sample. Enormous peaks of inorganic cations and commonly occurring basic amino acids can be seen in the electropherogram in Figure 3A. These substances are not completely separated because of overdosing the capillary for the analytes that are present in large contents in the blood plasma. To the contrary, this large stacking is necessary for submicromolar concentrations of methylarginines and the CE/C^4^D method is sufficiently sensitive for routine determinations of ADMA/SDMA in blood plasma (Figure 3B). Both the peaks of ADMA and SDMA are separated from the other analytes down to the C^4^D baseline (see the illustrative electropherograms of testing the method on several plasma samples obtained from various individuals, Figure 4). The determined plasmatic levels vary in the range 0.36–0.62 µM for ADMA and 0.32–0.70 µM for SDMA, which is in agreement with the levels measured by HPLC and reported in the literature [26]. The developed method is not suitable for the determination of MMA, whose plasmatic concentrations are an order lower than those of ADMA and SDMA. The MMA peak is overlapped by the peak of another analyte and it would be necessary to change the BGE composition or modify the whole method.

Calibration of the method was performed at six concentration levels in the range 0.1–4.0 µM, prepared by spiking a single plasma sample, which was then processed in the laboratory, see Table 1. The calibration dependences for the peak area are linear with a correlation coefficient of *R* > 0.999. The attained LOD values were 16 and 22 nmol/L of untreated plasma for ADMA and SDMA, respectively; the LOQ equalled 53 and 72 nmol/L of untreated plasma for ADMA and SDMA.

The repeatability of the determination of a single plasma sample calculated for ten subsequent determinations (the plasma sample was repeatedly processed) was 0.8–1.9% for the migration time and 2.1–3.8% for the peak area. The precision of the method, determined for three days, where the plasma sample was analysed 10 times every day, is 2.3–3.5% for the migration time and 3.7–5.2% for the peak area.

### 2.3. Comparison of Electrophoretic Determination of ADMA and SDMA

Table 2 provides a survey of the basis parameters of the electrophoretic determination of ADMA and SDMA. Six of the nine described methods for the determination of methylarginines in blood plasma were compared. Of these, the two CZE-LIF methods required conversion of the methylarginines into their fluorescently active FTIC or NDA derivatives using overly expensive off-line derivatisation [37,45]; SPE extraction of the plasma was used in case [45]. In contrast, the direct CZE-UV determination of methylarginines without derivatisation is connected with a complicated laboratory preparation of the plasma, based on single-step [41] or double-step [38] concentration of the sample (evaporation of the sample to dryness and its subsequent reconstitution). CZE-MS determination without derivatisation has low sensitivity where stacking is required for measuring the plasmatic levels [43]. In addition, the CE-MS combination is technically and financially demanding. Derivatisation, SPE and evaporation connected with reconstitution of the sample involve time-demanding and complicated operations, especially when a small amount of blood plasma is to be analysed.

C^4^D is a universal detection technique that does not require derivatisation. The laboratory preparation of the plasma for CE/C^4^D is simple and consists of deproteinisation of the sample by addition of ACN, and subsequent vortexing and centrifugation. The thus-prepared sample is injected directly into the capillary. In addition, only 15 µL of the plasma is required for the determination. The low LOD is fully comparable with the CE-LIF or CE-MS combinations and is achieved using stacking of a large volume of sample. Stacking takes place completely automatically without manual support and the individual steps are controlled by the software of the CE instrument, achieving high reproducibility of the determination. The developed method it thus suitable for monitoring ADMA and SDMA on large sets of samples.

## 3. Materials and Methods

### 3.1. Chemicals, Model Sample and Plasma Sample Preparation

All the employed chemicals are of analytical grade purity. N^G^,N^G^-dimethylarginine dihydrochloride (ADMA), N^G^,N^G^′-dimethyl-L-arginine di(p-hydroxyazobenzene-p′-sulfonate) salt (SDMA) were purchased from Sigma (USA); N^G^-methyl-L-arginine acetate salt from MMA, Fluka, USA; NaOH from Fluka, Buchs, Swiss; and acetonitrile (ACN) and acetic acid (AcOH, 99.8%) from Sigma-Aldrich (Steinheim, Germany). All the water solutions are prepared from deionized Milli-Q water (DEI, 18.2 MΩ cm, Millipore, France). The infusion solution for intravenous application is 154 mM NaCl in DEI from Ardeapharma (Czechia).

A model sample of methylarginines at a low concentration level is used for method development and its optimization; it had the following composition: 0.1 µmol/L equimolar mixtures of ADMA, SDMA and MMA were prepared from standard solutions of methylarginines at a level of 1 mg/L by dilution and addition of an infusion solution and ACN. The final content of the infusion solution in the mixture is 25% *v/v* and that of ACN is 75% *v/v* (250 µL/750 µL); this model sample composition simulates treatment of human plasma by deproteinisation using ACN in a ratio of 1:3 *v/v*.

Human plasma samples were obtained from healthy volunteers undergoing preventative examination and some residual plasma samples were provided by the Královské Vinohrady University Hospital (Prague, Czech Republic). The study was approved by the Ethical Committee of the Third Faculty of Medicine (Charles University, Prague, Czech Republic), head of review board Dr. Marek Vácha, PhD. Venous blood samples were collected in test tubes containing EDTA and, after initial treatment, the separated plasma samples were stored in a freezer at −80 °C until the analysis. Before the CE measurement, 15 µL of unfrozen plasma was mixed with 45 µL of ACN [51]. The mixture was vortexed (Vortex Genie 2, Scientific Instrument, USA) in a 200 µL Eppendorf tube for 3 min and then centrifuged under an acceleration force of 4000× *g* for 60 s (MiniSpin Plus personal microcentrifuge, Eppendorf, Hamburg, Germany), which is sufficient for complete precipitation of the plasmatic proteins. Subsequently, 50 µL of the supernatant was transferred to a plastic vial with an insert and subjected to CE analysis. For quantification, the unfrozen plasma was spiked with standard ADMA and SDMA in the concentration range 0.1 to 4 µmol/L, which was added directly to the untreated plasma. The thus-obtained calibration curves were subsequently treated by the addition of ACN, as described above.

### 3.2. Electrophoretic Apparatus

CE experiments were performed by using an HP^3D^ CE instrument (Agilent Technologies, Waldbronn, Germany) equipped with C^4^D originally developed by the ADMET company (Czech Republic). The C^4^D consists of two brass tubular electrodes, each 2.5 mm in length, with a 1.0-mm-long detection gap between them with inserted Faraday shielding. C^4^D operates with a sine-wave signal frequency of 1.0 MHz and 80 V effective voltage, which was received from a quartz crystal oscillator [52]. C^4^D is placed in the electrophoretic cassette together with a separation capillary. The capillary was pressurized from its long (injection) end. All the experiments were performed at a controlled temperature of 15 °C. For the C^4^D experiments, there was no need to make a window on the capillary. The electrophoretic resolution *R* was computed according to the equation, *R* = 2(*t*_M2_ − *t*_M1_)/(*w*_1_ + *w*_2_), where *w* is the peak width at the baseline, and *w* equals 1.7·*w*_1/2_. The LOD and LOQ values were determined from the peak height as the average concentrations corresponding to a signal/noise ratio of 3 or 10 (the background C^4^D noise is 5 µV).

### 3.3. PAMAPTAC-Coated Capillary

CE separation was performed in a fused silica capillary (Polymicro Technologies, Phoenix, USA) that was covalently coated by a copolymer of cationic 3-acrylamidopropyl trimethylammonium chloride (APTAC, 75% m/m solution in DEI, Aldrich) and acrylamide (Acros Organics), with a 6% molar content of APTAC in the polymerization mixture, further denoted as a 6% PAMAPTAC capillary, see Figure 5A. The capillary parameters are a 25 µm inner diameter (ID), 360 µm outer diameter (OD), total length of 39.45 cm and length to C^4^D 25.05 cm. The PAMAPTAC capillary was prepared in the laboratory according the procedure in [53,54] and described in detail in our recent publications [49,50]. The 6% PAMAPTAC capillary was characterised by an anodic EOF mobility with a value of (−17.3 ± 0.0) 10^−9^ m^2^V^−1^s^−1^ in 750 mM AcOH as background electrolyte (BGE). The EOF mobility was measured experimentally by the injection of a short DEI zone into the detection end of the capillary by application of negative pressure (−50 mbar) for 2 s. Subsequently, a high voltage of 30 kV without a ramp was applied and the migration time of water zone was monitored. Newly fabricated PAM and PAMAPTAC capillaries were flushed by BGE for 20 min only before the first separation.

### 3.4. Electrophoretic Stacking Coupled with Whole Capillary Filling by the Sample

In order to obtain a low LOD, electrophoretic stacking was developed. This was based on filling the whole capillary with the sample and subsequent electrophoretic sharpening of the sample by the effect of transient pseudo-isotachophoresis, originally described by Shihabi [55,56,57]. A description of the individual steps of the stacking procedure follows, see Figure 5B.

A vial with the sample is connected to the entrance into the capillary and a vial filled with BGE for the waste is connected to the exit from the capillary, and application of a pressure of 940 mbar for 2 min leads to filling of the whole capillary with the sample.The exit vial is exchanged for a vial with clean BGE and BGE with a length of 18.0 mm is forced to the exit end of the capillary by application of a vacuum impulse of −50 mbar for 2 min. The sample in the capillary has a length of 376.5 mm, corresponding to 95.4% of the overall length of the capillary.A vial containing BGE is placed at the entrance and the separation voltage of +30 kV, with a ramp of 0.2 min, is turned on. Methylarginines migrate in the cathodic direction towards the detection end of the capillary in the electric field. BGE is drawn out of the end of the electrophoretic vial into the capillary through the action of the anodic EOF and the EOF simultaneously forces the residual sample matrix out of the capillary into the entrance vial. Methylarginines are first isotachophoretically sharpened to the base of the long zone of the sodium ions (the most abundant cations in blood plasma) by the mechanism of transient pseudo-isotachophoresis [55,58], where sodium acts as leading ion and solvent ACN provide the high field strength necessary for analyte sharpening, similar to that provided by the terminating ion. Then, after the zone of sodium ions with the sharpened methylarginines moves out of the sample area into the BGE, the separation mechanism switches to the capillary zone electrophoresis regime (CZE) and the individual zones are separated.After the end of the separation, a vial with fresh BGE is connected to the entrance and a vial for waste is connected to the exit. The capillary is then rinsed by a pressure of 940 mbar for 5 min.

## 4. Conclusions

The CE/C^4^D combination provides a very strong instrument for the determination of ADMA and SDMA in blood plasma. A high sensitivity of the determination with LOD at the level of 10^−8^ M in the plasma is achieved by stacking a large volume of clinical sample, which fills 94.5% of the volume of the separation capillary. The residual sample matrix is forced out of the capillary during the stacking process by opposite-direction EOF, whose magnitude is controlled by covalent coating of the capillary with the cationic copolymer PAMAPTAC. The PAMAPTAC coating further prevents the adsorption of methylarginines on the capillary walls and enables the separation to be carried out in the counter-current regime. This results in a high electrophoretic resolution of the individual ADMA and SDMA isomers and their separation from the other plasma components. The determination is performed with only 15 µL of plasma, which is treated by adding ACN and subsequent elimination of the precipitated proteins.

## Figures and Tables

**Figure 1 ijms-22-02970-f001:**
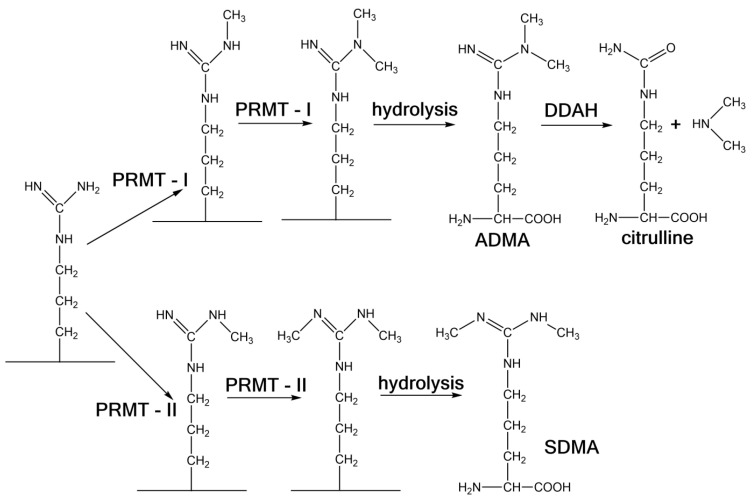
Double methylation of the side chains of arginine bonded to proteins through the effect of the protein arginine methyltransferases of Class I (PRMT-I) and Class II (PRMT-II), releasing ADMA and SDMA by protein hydrolysis; degradation of ADMA is through the effect of dimethyl arginine dimethylaminohydrolase (DDAH).

**Figure 2 ijms-22-02970-f002:**
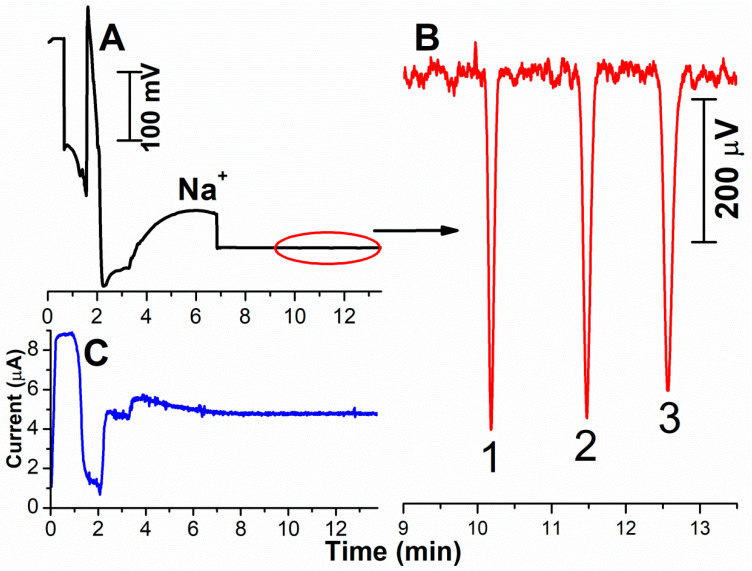
Complete recording of the stacking of the model sample (0.1 µM MMA, ADMA and SDMA in an ACN/infusion solution of 3:1), with filling 94.5% of the capillary with the sample (**A**); (**B**)—detailed recording of the separation of 0.1 µM methylarginines; (**C**)—variation of the electric current. Peak identification: 1—MMA; 2—ADMA; 3—SDMA.

**Figure 3 ijms-22-02970-f003:**
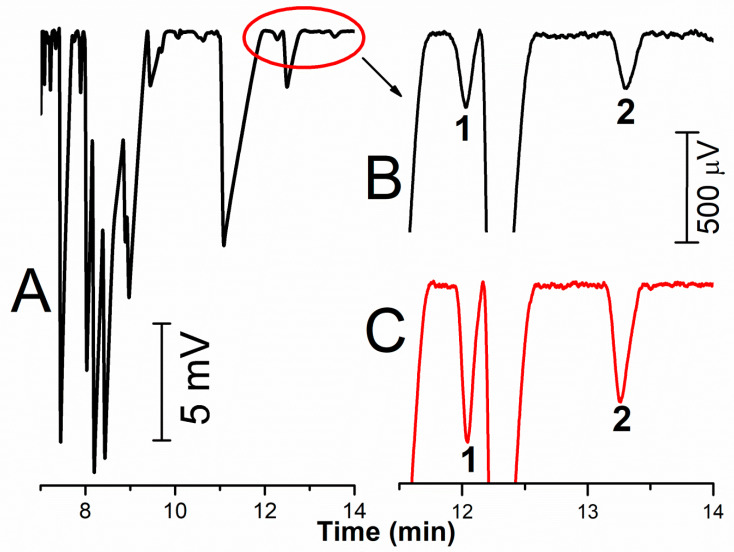
CE/C^4^D separation of human plasma treated by ACN (1:3) after filling the whole capillary with the sample (**A**); (**B**)—zoomed part of the electropherogram with the peaks of ADMA (1) and SDMA (2); (**C**)—the same plasma after the addition of 0.2 µM ADMA and SDMA.

**Figure 4 ijms-22-02970-f004:**
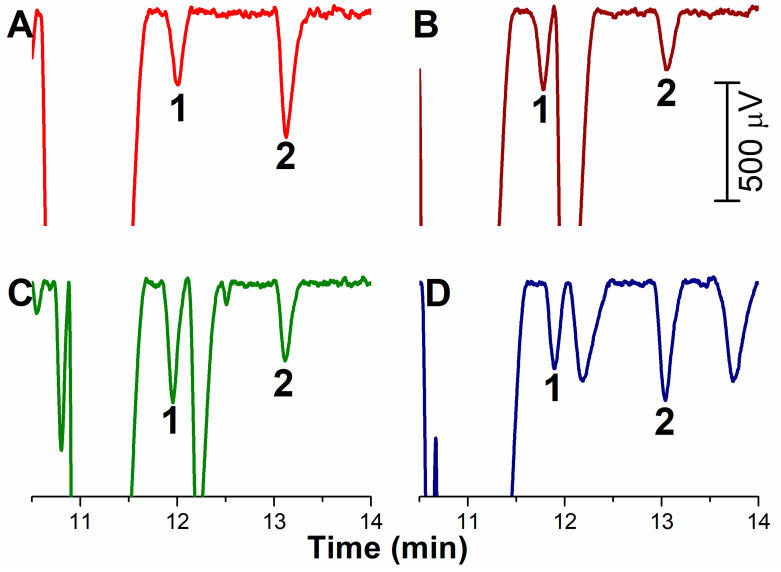
CE/C^4^D analysis of ADMA (1) and SDMA (2) in various samples of human plasma with concentrations determined as (**A**) 0.45 µM ADMA and 0.70 µM SDMA; (**B**) 0.36 µM ADMA and 0.32 µM SDMA; (**C**) 0.62 µM ADMA and 0.38 µM SDMA; (**D**) 0.44 µM ADMA and 0.70 µM SDMA.

**Figure 5 ijms-22-02970-f005:**
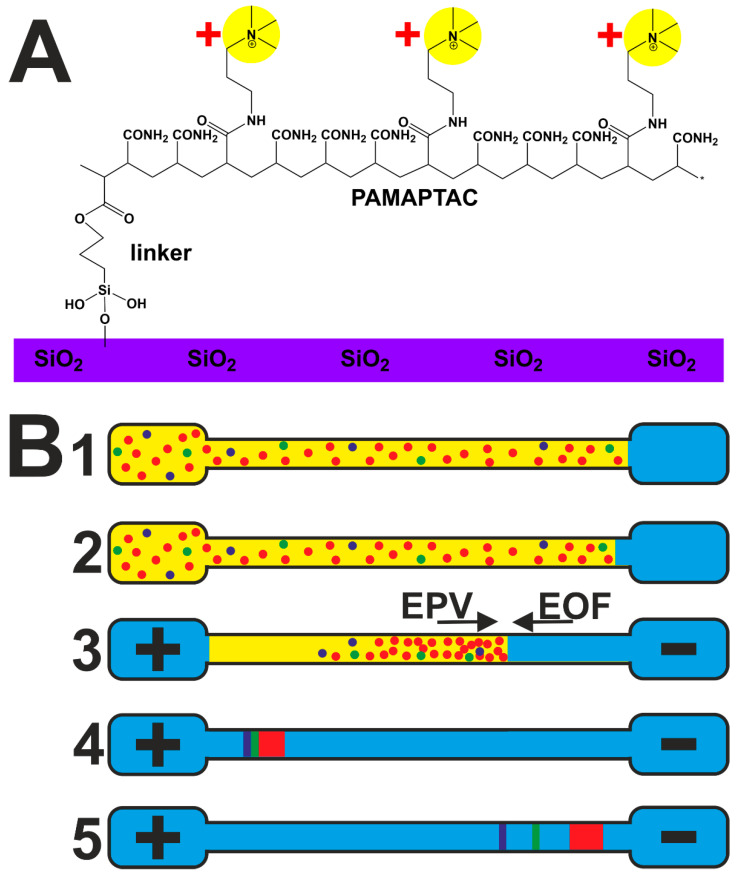
The copolymer PAMAPTAC covalently bonded through a linker (3-(hydroxysilyl)propyl methacrylate) to the inner surface of a fused silica capillary (**A**); (**B**) working procedure or electrophoretic stacking combined with the whole capillary sample injection: 1—filling the whole capillary through the application of a pressure of 940 mbar; 2—drawing short zones of BGE into the exit end of the capillary by an under pressure of −50 mbar; 3—turning on the separation voltage of +30 kV causes electrophoretic migration of the analyte (EPV—electrophoretic velocity) and drawing of the BGE from the detection end at the speed of the EOF; 4—stacking the analyte on the background of the zone of sodium ions by the transient pseudo-isotachophoresis technique; 5—separation of the concentration zones by the CZE technique. BGE—light blue; sample matrix—yellow; sodium—red; ADMA—green; SDMA—dark blue.

**Table 1 ijms-22-02970-t001:** Parameters of the calibration dependence for the determination of ADMA and SDMA in human plasma.

Parameter	ADMA	SDMA
Concentration range, µM	0.1–4.0	0.1–4.0
Slope—Area, Mv·s/µM	5.30 ± 0.05	5.86 ± 0.07
*R*	0.9997	0.9994
Slope—Height, mV/µM	0.95 ± 0.08	0.69 ± 0.08
LOD, nM	16	22
LOQ, nM	53	72

**Table 2 ijms-22-02970-t002:** Parameters of the electrophoretic and chromatographic determination of ADMA in the model samples and blood plasma, *t*_M_ migration or retention time.

Table 1000	Derivatisation	Matrix	Treatment	*t*_M_, min	LOD *, nM	LOD **, nM	Ref.
CE-LIF	FITC	plasma	deproteinisation, 1000-fold dilution	10	0.05	50	[37]
CZE-UV	-	plasma	deproteinisation, 2-fold evaporation	16	30	30	[38]
CZE-UV-stacking	-	plasma	deproteinisation, evaporation	22	10	10	[41]
CZE-LIF	NDA	model	-	8	20	-	[44]
MCE-LIF	NDA	model	-	3	70 ***	-	[44]
CZE-MS	-	model	-	11	500	-	[43]
CZE-MS-stacking	-	plasma	deproteinisation	14	20	20	[43]
CZE-LIF	NDA	plasma	heat-assisted SPE	15	5–8	5–8	[45]
CZE-C^4^D-stacking	-	plasma	deproteinisation	14	16–22	16–22	this paper
HPLC-FD	OPA	plasma	SPE, evaporation	30	10 ****	10 ****	[26]
HPLC-MS/MS	butyl ester	plasma	deproteinisation, evaporation	1.6	0.5	2 ****	[28]
GC-MS	pentafluoropropionyl	plasma	deproteinisation, evaporation	8.4	-	50 ****	[35]
ELISA	-	plasma	acylation, incubation	20 h	-	50 ****	[36]

* LOD in model sample; ** LOD in plasma; *** Estimate from the electropherogram; **** LOQ.

## Data Availability

Data is contained within the article.

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
