# Peer review of "Electrophoretic Determination of Symmetric and Asymmetric Dimethylarginine in Human Blood Plasma with Whole Capillary Sample Injection"

_ijms, 2021, doi:10.3390/ijms22062970_

Round 1

Reviewer 1 Report

The work is relevant and interesting. It is scientifically solid, with a detailed experimental procedure and clear results. Yet, I believe the following suggestions could help to increase its value. 

As the authors use HPLC as a comparison method. It would be interesting if they provide the results for HPLC as well. It would be also nice if they provide the results discriminated by patients as support information. 

A lot is discussed in the introduction about the time and complexity of different techniques. In that sense, it would convenient if table 2 was expanded to include the results of other methods, beyond CE, as well as the analysis time of each method. 

Please, include information about bioethical approval and donor consent in the M&M. 

Author Response

Dear Reviewer,

I would like to thank you for the very favourable evaluation of our paper. All the comments were fully taken into consideration and the manuscript has been amended accordingly. All the changes in the revised version are highlighted in red. I believe that the manuscript in the revised form will now be acceptable for publication in International Journal of Molecular Sciences.

  1. The determined concentrations of ADMA and SDMA in depicted electrophoregrams of human plasma are added to the caption of the Figure 5. In the next step, the developed methodology will be used for clinical monitoring of ADMA and SDMA in pediatric patients. We are still waiting for the delivery of the sample.
  2. Main chromatographic techniques and ELISA are newly included in the Table 2. A migration (retention) time is also added. The total analysis time cannot be determined from most articles, so retention time and sample treatment are included in the Table 2.
  3. Information about bioethical approval and donor consent are added to the paper.
  4. Comparative HPLC assays were not performed. It is only stated that the plasma levels of ADMA and SDMA in healthy individuals are in the range of the values determined by HPLC, which have been published in the literature.

With best regards,

Petr Tůma 

Reviewer 2 Report

The manuscript entitled ‘Electrophoretic Determination of Symmetric and Asymmetric Dimethylarginine in Human Blood Plasma with Whole Capillary Sample Injection’ describes a novel on-line sample preconcentration technique used for determination of methylarginines in human blood plasma by capillary electrophoresis coupled with contactless conductivity detector. The manuscript contains five figures and two tables.

Overall, the results presented in the manuscript are good and sufficient for publishing in International Journal of Molecular Sciences. In addition, a comparison with other CE methods reported for the determination of methylarginines makes it clear that the new approach presented in this manuscript has many advantages such as simplicity, short sample preparation and separation time, and low cost. However, there are some issues that need to be addressed prior to acceptance to publication.

1) There is not enough explanation of the stacking mechanism. The Authors suggest that the analytes’ focusing is caused by transient isotachophoresis (tITP) where sodium cations act as leading ions. Could Authors indicate which sample or background electrolyte components play the role of terminating ions?

Is it possible that the actual stacking mechanism is not the tITP but transient pseudo isotachophoresis induced by presence of acetonitrile in the sample matrix as reported by Shihabi (SHIHABI, Z.K., 2002. Transient pseudo-isotachophoresis for sample concentration in capillary electrophoresis. Electrophoresis, 23(11), pp. 1612-1617.)? If that is the case, the main difference between Authors’ method and that reported by Shihabi would be a sample matrix removal by EOF pump.

2) In the section 3.2 ‘Determining ADMA and SDMA in Human Plasma’ Authors said that ‘the developed stacking was applied to (…)’ (line 291). However, there is very little information provided about actual method development and/or optimisation (except for the statement in lines 225-228 ‘Separation of methylarginines as basic amino acids is performed in AcOH solutions. The tested concentration range varied from 250 to 2000 mM AcOH and the optimum separation of ADMA and SDMA from the other components of the plasma was obtained in 50 mM AcOH (pH 2.45), which was used as BGE in all the experiments described below.’ and another one in line 252-253 ‘A short BGE zone must be drawn into the capillary; otherwise the stacking results are worse’).

It think that it would be valuable to present some of those findings as a figure. Have Authors investigated any other parameters that might affect analytes’ focusing?

3) The main structure and the outline of the manuscript is well organised. However, the content of the third section (Results and Discussion) is a little chaotic. Foir example, Section 3.1. ‘Electrophoretic Stacking with Whole Capillary Sample Injection’ includes information about physic-chemical properties of the analytes, advantages of using C4D, and the description of stacking mechanism. Perhaps, some of that information might be moved to the Introduction Section (e.g., C4D discussion).

In addition, a use of complex sentences, often separated by semicolons makes it difficult to follow and might be confusing for the readers.

Author Response

Dear Reviewer,

I would like to thank you for the very favourable evaluation of our paper. All the comments were fully taken into consideration and the manuscript has been amended accordingly. All the changes in the revised version are highlighted in red. I believe that the manuscript in the revised form will now be acceptable for publication in International Journal of Molecular Sciences.

  1. I fully agree with you. This is the transient pseudo-isotachophoresis technique originally described by Shihabi. Sodium ions from blood plasma act as a leading ion and the organic solvent acetonitrile serves as pseudo-terminating ions. The correction is made throughout the article and new citations [55 – 57] are added.
  2. An additional electropherogram about development of the CE method is newly added to the Appendix A. CE separations of ADMA and SDMA were performed only in water solutions of acetic acid, which represent suitable background electrolytes for the determination of amino acids in combination with C4D. Other buffers have not been tested.
  3. The structure of the article is slightly modified. The description of the advantages and features of C4D is moved from Results and Discussion to the Introduction section. Also complex sentences, often separated by semicolons, are eliminated.

With best regards,

Petr Tůma